



# Intermittent heat instabilities in an air plume

Jean-Louis Le Mouël[1], Vladimir G. Kossobokov[1,2], Frederic Perrier[1], and Pierre Morat[1]

[1]Geomagnetism, Institut de Physique du Globe de Paris, 1 rue Jussieu, 75238 Paris, Cedex 05, France
[2]Institute of Earthquake Prediction Theory and Mathematical Geophysics, Russian Academy of Sciences, 84/32
5 Profsoyuznaya Street, 117997 Moscow, Russian Federation

*Correspondence to*: Vladimir G. Kossobokov (volodya@mitp.ru)

**Abstract.** We report the results of heating experiments carried in an abandoned limestone quarry close to Paris, in an isolated room of a volume of about 400 m³. A heat source made of a metallic resistor of power 100 W was installed on the floor of the room, at distance from the walls. High quality temperature sensors, with a response time of 20 s, were fixed on a 10 2-m long bar. In a series of 24-hour heating experiments the bar had been set up horizontally at different heights or vertically along the axis of the plume to record changes in temperature distribution with a sampling time varying from 20 s to 2 min. When taken in averages over 24 hours, the temperatures present the classical shape of steady state plumes, as described by classical models. On the contrary, the temperature time series show a rich dynamic plume flow with intermittent trains of oscillations, spatially coherent, of large amplitudes and a period around 400 s, separated by intervals of relative quiescence 15 whose duration can reach several hours. To our knowledge, no specific theory is available to explain this behavior, which appears to be chaotic interaction between a turbulent plume and a stratified environment. The observed behavior, with first order factorization of a smooth spatial function with a global temporal intermittent function, could be a universal feature of some turbulent plumes in geophysical environments.

## 1 Introduction

20 Thermal plumes, columns of hot fluid that rise above a localized heat source have received, like jets, a lot of attention (Turner, 1973). Numerous processes in the Earth and in small-scale environmental sites require the description of the effect of injecting heat or matter from natural and/or industrial sources into a stationary organized system, such as the ocean, the lake, or the atmosphere (Woods, 2010).

25 Models of plumes (and jets) have existed for decades. Models rely on a turbulent entrainment of ambient fluid in a shear layer within the edges of the plume, and a hypothesis of complete similarity along the downstream axis OZ of the plume. In these contributions, the entrainment rate of ambient fluid is proportional to the vertical velocity $w$ (along OZ), with the same entrainment constant $\alpha_e$ along $Z$. Such is the case of the classical model of Morton et al. (1956) who used three conservation equations (for fluxes of mass, momentum and buoyancy) to get the expressions of the temperature difference, plume radius, 30 and mean velocity along $Z$, $w(z)$. Fisher et al. (1979) got the same expressions using essentially dimensional analysis. In fact,



this hypothesis of complete similarity is strong and debated, not supported by all experiments. Recently, Crouzeix et al. (2003) resumed the study of the similarity, using available experimental data and concluded in favor of local states of partial self-similarity, in accordance with the theoretical analysis by George (1989), and evolving along the Z coordinate according to a universal route. Thermal plumes, columns of hot fluid that rise above a localized heat source have received, like jets, a

5 lot of attention (Turner, 1973). Numerous processes in the Earth and in small-scale environmental sites require the description of the effect of injecting heat or matter from natural and/or industrial sources into a stationary organized system, such as the ocean, the lake, or the atmosphere (Woods, 2010).

All those models are for stationary, time independent jets or plumes. In the present paper, we will turn to temporal fluctuations within the plume and in its vicinity, and reveal a dynamically chaotic plume. Our sampling in time and space,

however limited, allows us to describe the main features of this dynamic flow, whose observation, to our knowledge, is unprecedented.

## 2 The heating experiments in the Vincennes quarry

### 2.1 The Vincennes quarry

Various thermodynamic experiments have been carried out from February 2002 to April 2005 in an abandoned quarry

located in Vincennes, close to Paris (Crouzeix et al., 2003). The Vincennes quarry has a surface of 32,000 $m^2$, its ceiling is at a depth varying from 14 to 20 m. The walls show a section of the different tabular layers that have been exploited in a Lutetian formation, mostly limestone beds. The quarry consists of corridors and rooms separated by pillars of different sizes, most of them supporting the roof. Room shapes are roughly rectangular; the height of their ceiling varies between 1.5 and 7 m (the distance from the ceiling to the surface of fillings, about 2 m thick covering the rock floor). The total air volume in

the quarry, difficult to evaluate precisely because of unexplored and collapsed sections, is estimated to be around 80 000 $m^3$. The quarry is connected to the ground surface by a single large access pit with 4.56 m diameter (Perrier et al., 2002; Perrier et al., 2005; Perrier and Le Mouël, 2016).

Temperature had been measured in the quarry since 2001, giving a mean value of 12°C in 2003 and afterwards an increase of about 0.1°C per year. An annual variation of the order of 0.8°C is related to natural ventilation through the access pit (Perrier

et al., 2004; Perrier and Richon, 2010), which is active in winter times. At times smaller than a day, temperature variations are mostly due to variations of atmospheric pressure (Perrier et al., 2001), and do not exceed 0.03°C peak to peak in the absence of perturbations (visits, heating experiments). As in most underground systems, relative humidity in the atmosphere of the quarry is high (99.2 – 99.8 %).

Among various observations of temperature, we took advantage of the exceptional conditions of stability offered by such a

30 large quarry to conduct a long series of measurements of temperature on a plume set up in an isolated room of a volume of about 400 $m^3$. This set of long duration heating experiments is the subject of the present paper.



The room S15 selected for the heating experiments has a surface of about 24 m by 9 m (Fig. 1) with an average height of about 2 m (Fig. 2). Air exchanges with the rest of the quarry proceed through two entrances of about 3 and 3.5 m wide (Fig. 1). In order to reduce ventilation effects, the experiments were performed in the inner part of the room at distance from the entrances. On July 2nd, 2003, this part of about 12 m by 9 m in surface had been insulated with a wall made of Styrofoam.

However, most experiments reported in this paper were performed before the installation of the partition wall. Nevertheless, the natural ventilation of the section of room S15 used for the experiments is small, with a ventilation rate estimated to be of the order of $2 \times 10^{-6}$ s$^{-1}$ (Perrier and Richon, 2010).

## 2.2 Temperature measurements. Thermistors. Calibration.

We will consider space and time variations of the temperature with an amplitude as small as 0.01°C, even a few 0.001°C; so

we need very sensitive sensors. For this reason, and because the experiment relies on temperature data, we will briefly describe the sensors and their calibration. The three types of temperature sensors commonly used are thermocouples, resistance temperature detectors (RTD), and thermistors. Thermistors are very sensitive, cheap (one can buy and use many of them), and present a weak noise. However, their response time can be somewhat long, and must be known; they are not very stable on long term, but we are considering relatively short time constant relative variations (see below). A thermistor is

made of semi-conductor materials whose electric resistance $R$ decreases monotonously when the temperature increases. The relation between temperature $T$ and $R$ is strongly non-linear, often written in the form, $T^{-1} = A + B \, lnR + C \, (lnR)^2$, where $A$, $B$, and $C$ are coefficients to be determined. In the present study, we measure temperature space and time variations smaller than a few degrees, simultaneously at ten or twelve locations (see section II.3). What we need is an accurate inter-calibration of n thermistors, or $n$ thermistors plus a unique reference one. In other words, the relative calibration of a series of n

thermistors consists in reducing the value of the resistance $R_i$ ($i$ = 1, 2, … , $n$) of thermistor to the value $R_{ref}$ of the reference thermistor at the same temperature. For that all the thermistors are plunged in a bath of uniform temperature, controlled by a thermometer with 0.001°C accuracy. From a number of measurements in the bath, we determine the coefficients of the simplified relationship:

$$R_{ref} = a_i + b_i \, R_i \, , \quad i = 1, 2, \dots , n \qquad (1)$$

As a result, the temperature variations measured in the experiments, at all locations i, after using relation (1) are the same as if measured by the reference thermistor within 0.005°C (Crouzeix et al., 2003). The response time of the thermistors is in the range of 15-20 s. The sampling interval of the records is 20, or 40, or 120 s (Table 1).

## 2.3 Configuration of the measurement device of the experiment

Four set-ups instrumented with ten thermistors each were installed in a part of the room at distance from entrances. The

sensors, 30 mm long and about 5 mm in diameter, had an intrinsic response time of about 20 s. The precision of relative temperature measurements was found to be about 0.005 °C (Crouzeix et al., 2003). One of the set-ups, AIII, was placed just above the heat source and, unlike the other three, which remained at the same positions, was configured differently in



thirteen heating experiments from March 10 to June 9. Specifically, the 2-m bar with thermistors was placed vertically in experiments A1-A3 to evaluate the effect of the source at different heights, and that of the screen, then horizontally in experiments A4-A7 successively at a height of 1, 2, and 1.5 m (see Fig.2 and Table 1). Moreover, the sampling rate of measurements in AIII varied from once in 20 s in the first four experiments to once in 120 s in the last two. The set-ups AI and AII were used to measure the temperature in vertical air columns away from the walls, while the sensors of set-up S were placed at about 1.5 cm inside the rock of the ceiling and wall and in the filling of the floor. These later experiments, dedicated to studying the reaction of the room to the heating, are outside the scope of the present paper, where we focus on the experiments studying the plume itself.

Most of the heating experiments (including all considered in the present study) lasted for 24 hours, from Monday 12 a.m. to Tuesday 12 a.m. and were separated from the previous one by six days. The heat source was a metallic resistor of power 100 W with a rectangular surface of 5 by 7 cm and height of 15 cm. Measurements in the laboratory have shown that the source reaches its maximum temperature within 10 minutes. During the experiments, it was located about 3.4 m from the nearest back wall of the room (Fig.1). Except for one experiment (A2, see Table 1), the source was put directly on the floor as indicated in Fig.2. To study the possible contribution of sideway radiations, a screen was placed around the source in experiments A3, A5-A7 (Table 1).

## 3 Results: Temperature fluctuations in the plume

We now present the data which are at the core of the present paper. The temperature variations are measured along the two meters long bar AIII, by sensors ch1, ch2, … , ch10. In experiments A5a, A5b, A7a, A7b, A6a, and A6b the bar is maintained horizontally, centred above the heating source, at a height of 1 m for A5, 1.5 m for A7, and 2 m for A6 above the floor (Table 1). In experiment A3 the bar is maintained along the vertical of the source. These experiments were all performed with a screen around the source, in order to cancel or reduce radiative heating.

A previous study of the thermal stratification induced in the same room S15 by the heating (Crouzeix et al., 2006) had shown that temperature variations in the environment outside the plume were small compared to temperature variations in the plume itself; we will consider that the plume is in an environment with a uniform temperature $T_0$. In the following graphs the temperatures are reckoned from the reference temperature of the environment $T_0$, $\Delta T(t) = T(t) - T_0$.

### 3.1 The horizontal recordings at 1.00 m

Fig. 3 shows the recordings of temperature at a height of 1 m, made from 12:00 on April 7 to 16:00 on April 8, 2003 during and 4 hours after the heating experiment A5a, using a sampling interval of 40 seconds. After a transient phase of about an hour from switching on the heating source, the curves take the form of regular spikes of high amplitude at sensor 5 (up to 1.5°C), smaller but still high on sensors 4 and 6. Note that the curves 4 and 6 are not always exactly affine one of each other,



which is presumably an effect of shifts of the plume, and, in fact, present with smaller and smaller amplitudes on all the curves.

The duration of the temperature peaks of instability is close to six minutes. They appear either isolated, or in pairs, or in trains; when a peak is isolated, or is the last of a train, the declining phase of instability is longer than 6 minutes. Between these temperature instabilities phases of stability are present, sometimes longer than 5 hours.

Fig. 4 displays the temperature data from the same experiment in the form of a color-coded contour space-time map (level lines) over the time interval from 2003/04/07 9:00 to 2003/04/08 15:00. The colour scale indicates the amplitude of the measured temperatures. We observe more vividly the trains of nearly regular strong heat pulses along with their spread along the horizontal device. These heat pulses of instability should not be taken as physical drops of air observed at a given time.

Let us zoom on the train of instabilities of April 7, from 18:00 to 20:00 (Fig. 5, upper plate). The peaks take the form of localized pulses, each about 380 seconds. Look at the abrupt shift of the plume at about 18:30. Another illustration is coming from the same experiment A5a from 4:00 to at 6:00 on April 8 (Fig. 5, lower plate). One can see six isolated pulses with peak to peak inter-event times of 420, 480, 840, 960, and 1360 s (which are suggestive of a likely "doubling of period") followed by a train of at least six of apparently connected pulses whose mean duration is about 320 seconds.

### 3.2 The horizontal recordings at 1.50 m

A similar analysis of temperatures collected at the height of 1.50 m has been performed (experiments A7a on May 26-27 and A7b on June 2-3, 2003). Fig. 6 shows the recordings from 11:00 on May 26 to 17:00 on May 27. Clearly, the general behavior of the recording is the same as at the height of 1.00 m (section III.1), although the larger sampling rate of one per 120 seconds provides coarser curves. Nevertheless, the contrast between active and quiet segments (as well as duration of the phases of stability) is smaller than in the case of temperatures at 1 m height.

A zoom on a train of the temperature peaks, in space-time representation, is illustrated by Fig. 7. The mean peak to peak time for the eight pulses of the train starting just before midnight (at about 23:58) is 8 minutes (± 40 s). However, the individual durations of pulses range from 360 up to about 600 seconds, which might be considered as a mixture of single and double periods; or the first and the last heat pulses might be considered as isolated ones. We point out again that in 2003 we were not able to make simultaneous recordings, i.e. to perform a direct comparison of several temperature recordings made at the same time at several positions above the source.

### 3.3 The horizontal recordings at 2.00 m

Fig. 8 shows the results of experiment A6a made on April 21-22 at the height of 2 m. Again the recording presents periods with and without peaks, with the largest amplitude at sensors 4 and 5 (in fact, very similar). One observes a slow temperature increase from 13:00 to 19:53 on April 21, then a descent till 22:13, followed by a flat behavior till the end of the heating phase at 12:00 on April 22. At this 2-meter height, the interaction of the plume with the ceiling makes the situation somewhat more complex. Fig. 9 shows two zooms on the time intervals 18:53 – 20:33 on April 21 and 4:11 – 5:51 on April



22. Despite the 120-second sampling, it is seen that, when the oscillations are large enough, they are practically in phase at all sensors, i.e. the temperature varies in the same way in the whole plume. In the first interval we observe a train of ten temperature pulses with duration of about 410 s each, while in the second interval we observe two trains of five and four pulses, respectively, with durations of about 440 and 425 s respectively. Fig. 10 gives a representation of a long train of ten

pulses from 19:12 to 20:20 on April 21; their mean duration is 6.8 minutes (± 34 s) with a larger variability (from 240 to 600 s). Moreover, we observe a companion series of instability pulses of heat separated from the main one.

### 3.4 AIII device maintained vertically

In experiments A1, A2, and A3 (Table 1), the device AIII is held vertically, with sensor ch10 close to the heating source (placed on the floor). Results of experiment A3 are displayed in Fig. 11, in the form of space-time presentation of the

temperature level lines (as in Figs. 4, 5, 7, and 10). At each moment of time $t_k$ we report the value of temperature $T(h_i, t_k)$, $h_i$ being the height at location of the $i$-th sensor ($i$=1, 2, …, 10); $t_k = t_0 + 20k$ in seconds. A series of conspicuous dilatations or contractions of the $T$ sections appear all along the graph; in general, the dilatations are sharp, the following contractions much slower. Unfortunately, no simultaneous horizontal recording exists (a single AIII device was available). Fig. 12 shows a zoom of Fig. 11, 1 hour 23 minutes long.

### 3.5 Joint study of the various time series

The intermittent nature of the various temperature signals is further studied in Fig. 13. On the left side of the four plates in Fig. 13, the times of the 50 largest local maxima of the average temperature of all the sensors on the AIII bar are shown for each of the three experiments performed at height 1, 1.5, and 2 m (A5a, A7a, A6a) in horizontal position and the experiment (A3) in vertical position. The empirical distribution function of the obtained 49 inter-event times, expressed in fractions of a

20   day, is plotted on the right side of each plate. Table 2 summaries a few statistics of these inter-event times, when split into two classes of short and long intervals by the unique threshold of 12 minutes. Except for the bar in vertical position (first line), the mean of the short times ranges from about 6 to 8¼ minutes, while the mean of the long times is about 5 or more times larger with the largest ones lasting for a few hours. Overall, the statistics of the inter-event times is remarkably similar in all experiments.

## 4 Averaged temperature profiles

Let us consider now in brief the average temperatures $T_h$ at height $h$ taken over the 24 hours of each heating experiment. Vertical profiles (Fig. 14a) present a negative gradient $\Delta T_h = T_h - T_0$ in the lower part of the room, which turns positive in a layer about 50 cm thick below the ceiling. On the horizontal profiles (Fig. 14b), maximal averaged temperatures are observed on sensors 4 and 5, although the heat source is located below sensor 6. As expected, the width of the plume is

getting wider when the AIII bar is placed higher, and the temperature profile presents the classical bell-shape form observed





in thermal plumes generated by a point source. A Gaussian form of the temperature radial profile $T(r)$ in the plume (where $r$ is the radial coordinate) is often presupposed in stationary plume models, which focus on the variation of mean width, velocity, temperature, versus the vertical coordinate (Morton et al., 1956; Landau and Lifshitz, 1987).

5  We will just make a few comments on our observations in the light of stationary models. Plumes belong to the class of free convection flows, maintained by temperature differences. "No velocity scale is provided by the specification of a free convection situation". Nevertheless, an estimate of the order of magnitude $U$ of the velocity can be obtained, with some caution, from the Navier-Stokes equation for the steady state:

$$\vec{u}\overrightarrow{\nabla u} = -\frac{1}{\rho}\overrightarrow{\nabla \rho} + \nu\nabla^2\vec{u} - g\alpha\,\Delta T\vec{z} \qquad (2)$$

where $\vec{u}$ is the velocity, $\vec{z}$ the upward vertical unit vector, $\rho$ is the air density, $\nu$ its kinematic viscosity ($0.15\ 10^{-4}\ \mathrm{m^2 s^{-1}}$), $g$ the

10  gravity acceleration ($\sim 10\ \mathrm{ms^{-2}}$), $\alpha$ the expansion coefficient ($\alpha \sim T^{-1} \sim 3\ 10^{-3}\ \mathrm{K^{-1}}$), $\Delta T$ the temperature anomaly taken from Fig. 14b as 0.3 °C, the buoyancy $g\alpha\,\Delta T \sim 10^{-2}\ \mathrm{ms^{-1}}$. From $\vec{u}\overrightarrow{\nabla u} \sim U^2/L \sim g\alpha\,\Delta T$, $U \sim (g\alpha\,\Delta TL)^{1/2}$, which gives $U \sim 10^{-1}\ \mathrm{ms^{-1}}$.

For this estimate to be valid, we have to check *a posteriori* that the viscous force is weak compared to the inertial one: $\vec{u}\overrightarrow{\nabla u}$ / $\nu\nabla^2\vec{u} \sim UL / \nu \sim (g\alpha\,\Delta T\,L^3 / \nu^2)^{1/2} = \mathrm{Gr}^{1/2} \sim 2^{-1/2}\,10^4$; Gr is the Grashof number which for in our case of air has a large value. The flux $F$ of the buoyancy $B = g\Theta = g\alpha\,\Delta T$ at height $z$ is:

$$F(z) = g \int_{S(z)} w(z)\,\Theta\,(z)\,ds \qquad (3)$$

$w$ is the vertical mean velocity, S the area of the section of the plume. Let us take $z = 1$ m, $w \approx U \approx 10^{-1}\ \mathrm{ms^{-1}}$, $S(1\ \mathrm{m}) \approx 10^{-2}$ $\mathrm{m^2}$; from Fig. 3b the mean of $\Theta$ in the section of the plume at the 1-meter altitude is $\int_S \Theta\,(1\ \mathrm{m})\,ds \approx 1.43\ 10^{-3}\ \mathrm{m^2}$. Then,

$$F_{z=1\,\mathrm{m}} \approx 1.4\ 10^{-3}\ \mathrm{m^4\ s^{-3}} \qquad (4)$$

Let us compare this estimate of $F$ with the flow of buoyancy delivered by the source:

$$F_0 = g\,\alpha\,P\,(\rho\,C_p)^{-1} \qquad (5)$$

$P$ is the power of the source (100 W), $\rho$ the air density at 12.5°C ($1.236\ \mathrm{kg\ m^{-3}}$), $C_p$ the specific heat of air at constant pressure ($C_p = 1006\ \mathrm{J\ kg^{-1}\ K^{-1}}$). It comes

$$F_0 = 2.4\ 10^{-3}\ \mathrm{m^4 s^{-3}} \qquad (6)$$

The comparison between $F_0$ and our estimate of $F$ can be judged satisfactory.

25  *Remark:* Assuming a Gaussian shape for the horizontal profiles of temperature difference $\Delta T(r,z) \sim \exp(-r^2/b^2(z))$, $r$ being the radial distance from the maximum of $\Delta T(z)$ in the planes $z = 1$, 1.5, and 2 m, the "radius" $b(z)$ can be estimated from the measurements; it is found that this radius $b(z)$ is larger and the temperature difference $\Delta T(z)$ is several times lower than the values predicted by the Morton et al. (1956) model. Horizontal profiles show that the mean plume is deviated from the vertical of the source; it is likely that it also deviates in the perpendicular horizontal direction (no data). Measured





temperatures should then be corrected before being compared with model predictions. Phase changes of water can also be a cause for temperature being lower than predicted by the model, as well as some inadequacy of the model.

## 5 Some global characteristics of the plume dynamics

We presented above (section IV) the average temperature profiles obtained from our heating experiments as a representation of stationary plume models, which raises some interesting questions (which will be touched upon below). The interest and novelty of our study, however, relies in the higher frequency content of our time series of temperatures in the plume (Figs. 3-13). Let us thus come back to the plume dynamics.

Consider $\bar{T}(t)$ being the average temperature recorded over the ten sensors on the bar. Fig. 15 shows the energy spectra of the series of the empirical first derivative of the average temperature, $(\Delta\bar{T}(t))/\Delta t$, in experiments A6a and A6c at 2 m height (top), A7a and A7b both at 1.5 m height (middle), and A5a and A5b both at 1 m height (bottom). The maximum peaks appear at periods of about 478 and 467, 474 and 388, 395 and 467 seconds respectively; the average is 445 seconds and the standard deviation 42 seconds.

From the comparison of the temperature variations $\Delta T(t)$ registered along the horizontal bar AIII disposed at the three heights of 1, 1.5, and 2 meters, it appears that (as already pointed out above) the curves $\Delta T_i(t)$ relative to the different sensors $(i = 1, 2, \ldots, 10)$, to a first approximation, are affine to each other:

$$\Delta T_i(t) = a_{ij}\,\Delta T_j(t) \qquad (7)$$

For example, taking $j = 5$ of the sensor seeing the large variations, $a_{ij}$ being (constants) parameters

$$\Delta T_i(t) \approx a_{i5}\,\Delta T_5(t) = b\,\Delta T_5(t) \qquad (8)$$

This observation is not trivial. First, it clearly confirms the significance of variations of a few hundredths of a degree and the quality of the calibration. Second, and more important, it demonstrates that the plume varies grossly "en bloc" in time. In other words, extrapolating a bit boldly, we have:

$$\Delta T_z(r,t) \approx f_z(r)\,\theta(t) \qquad (9)$$

$\Delta T_z$ is, at each $z$, the product of a space function $f_z(r)$ by a time function $\theta(t)$. Making a step further, we assimilate $f_z(r)$ to $\exp(-b^2/r^2)$, $b$ depending on $z$ according to a self-similar law. Note again that we have available only three altitudes $z$ (1.0, 1.5, and 2 m, respectively), and no simultaneous recording at two altitudes.

The causes of the observed instabilities remain unclear. The mean axis of the plume could be affected by unstable motions and distortions, especially given the fact that external influences cannot be completely ruled out before the construction of the partition wall (Fig. 1). Complementary experiments performed after the completion of the wall, however, indicated that the mean barycenter of the plume can indeed move, but that this effect is not dominating, and that the observed instabilities of the plume are not a consequence of slight plume motion, but large and intrinsic instabilities of the plume itself.



## 6 Discussion and conclusions

In this paper, we report on the temporal behavior of turbulent plumes in geophysical conditions, which still remains poorly known and rarely studied. The instabilities observed in our heating experiments, nevertheless, seem to be a universal and

familiar feature. For example, in experiments simulating plumes with saline solutions in large tanks, the recordings reveal the peculiar temporal instabilities of the plumes, with the occurrence of transient large voids, even close to the plume axis, and also rather high above the plume source (Fig. 16).

Our experiments and our data sets, while able to reveal important aspects of the instabilities, nevertheless, suffer from a number of limitations. First, we made only temperature measurements. Second, the ten sensors were attached every 20 cm on

a 2-m long bar. Because we had only one moveable bar, no simultaneous recordings for different arrangements are available. Third, air velocity was not simultaneously measured and our temperature measurements could not be transferred into estimates of velocity. Fourth, the response time of our sensors (20 s) did not allow us to access the probably important higher frequency part of the temporal effects. Finally, the experimental room was subject to a small level of natural ventilation during our experiments, and an influence on the dynamical regimes of the plumes cannot be completely ruled out.

Despite of that, our experiments indicate that, in a first approximation, the temperatures $\Delta T(x,t)$ recorded versus time $t$ along the horizontal bar, at various positions $x$ above the source, suggest essentially spatially coherent trains of pulses arranged in a quasi-periodic manner, with durations of 360-400 s, separated by intervals of stability, which can last up to several hours. The local response function $\Delta T(x,t)$ thus appears as the product of a smooth spatial function $S(x)$ by a non-linear mechanism $F(t)$, generating a chaotic regime (note e.g. that doubling of period are observed). It would be interesting to explain the value

of the periods observed during quasi-periodic regimes, and that of the intermittent intervals of stability (up to several hours). Clearly, the observed factorization of $\Delta T(x,t)$ function is reminiscent of chaotic solutions of a system of non-linear differential equations mimicking the behavior of, sometimes complex actual dynamic systems. As an illustration, one may think of one coordinate of the celebrated system of the Lorenz equations (Lorenz, 1963). Such an exercise however cannot really be attempted for the dynamical plume, due to the limitations of our observations and additional important aspects of

the problem. Indeed, the dynamic plume is the major acting ingredient of a filling box with non-adiabatic boundaries (Linden, 1990; Crouzeix et al., 2006). During a 24 hours experiment, heat is accumulated at the ceiling and exchanged with the ceiling and side walls, causing probably evaporation and condensation, and significant associated heat transport by phase changes of water. The hot air, thus cooled at the ceiling is then fed again into the plume, with a circulation time which is probably an important characteristic time of the plume dynamics, and contributing to the intermittent quasi-periodicity.

Nevertheless, the plume integrates the complexity of the various heat relaxation scheme and feedback into an overall simple organization, with a first order factorization of the spatial and temporal variations. While numerical simulations have started to shed light on these mechanisms (Hernandez, 2015), experiments remain necessary to establish important properties.



Some aspects of this dynamics could be captured in the experiments performed in water tanks alluded to above (Fig. 16), and also point to the intermittent behavior of the reversals of the mean wind in Rayleigh-Bénard convection (Sreenivasan et al., 2002; Sugiyama et al., 2010). In the case of Rayleigh-Bénard convection, chaotic temporal variations arise despite the constant boundary conditions, in the absence of a source of motion, for a non-localized source of buoyancy. In the case of

plumes in confined environments, the plume or jet itself is a source of velocity in addition to buoyancy (Hernandez et al., 2015; Lopez and Marques, 2013). Despite the variability of conditions, the observed dynamical behavior seems to be remarkably similar to the behavior of the flickering candle (Maxworthy, 1999), and could be considered, tentatively, as universal. However, the apparent similarity of the situations may be due to the limited spatial sampling in our experiment. More elaborated and dedicated experiments are needed to study the temporal variations inside a turbulent plume and also in

its environment. In confined situations, indeed, the plume dynamics might result from the interactions of the plume with its environment and the various relaxation times that it can provide. Underground environments offer a promising context where these poorly known aspects of fundamental physics could be studied fruitfully, potentially providing useful insights for situations of geophysical or industrial relevance.

**Acknowledgement**

The authors acknowledge Catherine Crouzeix for the invaluable unique collection of data recorded in an underground quarry in Vincennes.

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





## Tables

**Table 1: Configurations of the experiments carried out in March-June 2003. Each was performed with a heat source of 100 W and lasted 24 hours. AIII position indicates the device orientation: V for vertical and $H_z$ for horizontal at altitude z.**

| Name | Date | AIII position | Source raised | Screen | Sampling time |
|------|------|---------------|---------------|--------|---------------|
| A1 | 03/10 | V | - | - | 20 s |
| A2 | 03/17 | V | + | - | 20 s |
| A3 | 03/24 | V | - | + | 20 s |
| A4 | 03/31 | $H_1$ | - | - | 20 s |
| A5 | 04/07, 04/14 | $H_1$ | - | + | 40 s |
| A6 | 04/21, 04/28, 05/05, 05/12, 05/19 | $H_2$ | - | + | 120 s |
| A7 | 05/26, 06/02 | $H_{1.5}$ | - | + | 120 |

**Table 2. Peak to peak inter-event time statistics: ID identifies the 24-hour heating experiment; AIII indicates position of the 2-m**

10 **bar: V for vertical and $H_z$ for horizontal at height z; $N_s$ is the number of short ($\Delta t < 12$ minutes) intervals; $N_l$ is the number of long ($\Delta t > 12$ minutes) intervals; $E(\Delta t_s)$ is the average duration of the short intervals, in $s$; $E(\Delta t_l)$ is the average duration of the long intervals, in $s$; columns n1, n2, and n3+ give the number of single, double, and multiple pulses, correspondingly.**

| ID | AIII | $N_s$ | $N_l$ | $E(\Delta t_s)$, $s$ | $E(\Delta t_l)$, $s$ | $\max(\Delta t)$, $s$ | n1 | n2 | n3+ |
|----|------|-------|-------|---------------------|---------------------|----------------------|----|----|-----|
| A3 | V | 26 | 23 | 227 | 3421 | 18840 | 15 | 3 | 6 |
| A5a | $H_1$ | 29 | 20 | 357 | 3748 | 17560 | 12 | 5 | 7 |
| A5b | $H_1$ | 22 | 27 | 395 | 2801 | 20840 | 17 | 10 | 4 |
| A7a | $H_{1.5}$ | 24 | 25 | 495 | 2400 | 7680 | 20 | 10 | 2 |
| A7b | $H_{1.5}$ | 25 | 24 | 432 | 2815 | 11280 | 19 | 4 | 6 |
| A6a | $H_2$ | 30 | 19 | 432 | 2185 | 6360 | 10 | 5 | 7 |
| A6c | $H_2$ | 31 | 18 | 449 | 2473 | 7560 | 13 | 4 | 6 |





## Figures

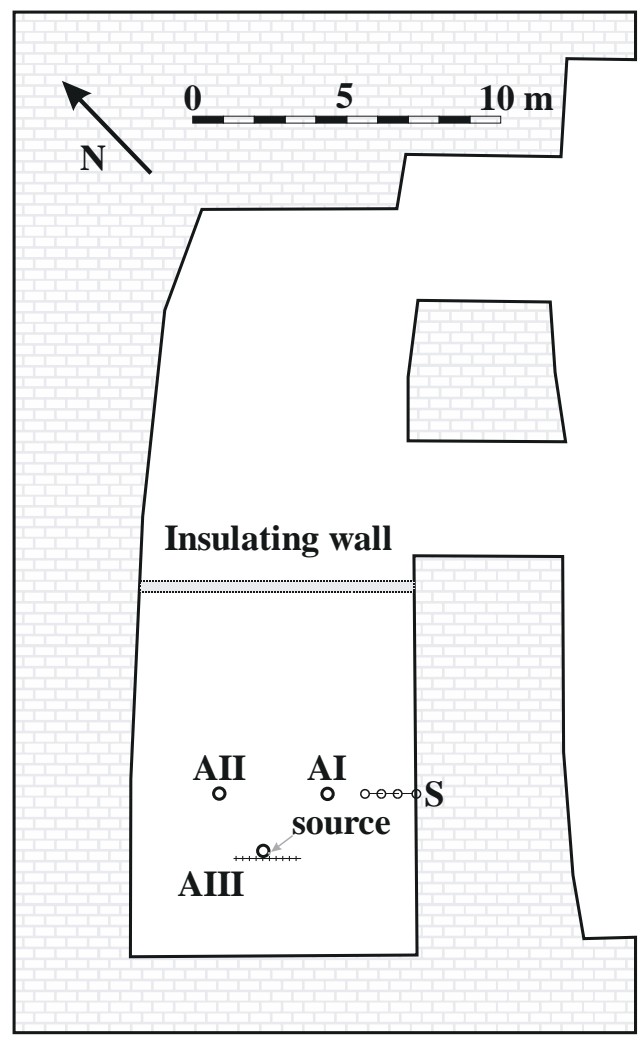

5 **Figure 1: The Sketch of the experiment room S15 in the Vincennes quarry showing the location of temperature set-ups AI, AII, AIII, and S, and of the insulating wall installed in July 2003.**




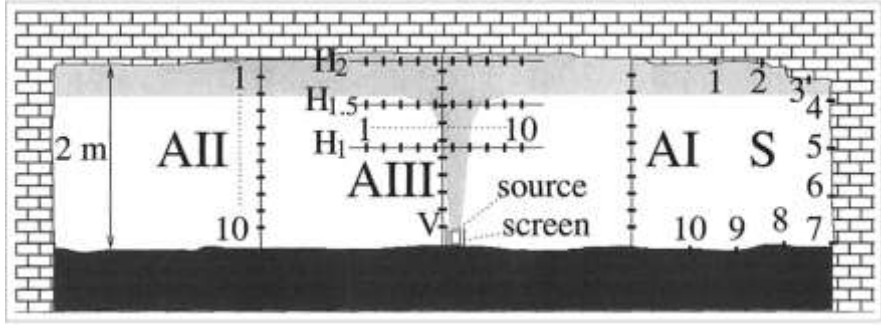

**Figure 2: Schematic cross-section of the experimental room S15 showing the four set-ups along with the positions of the temperature sensors.**

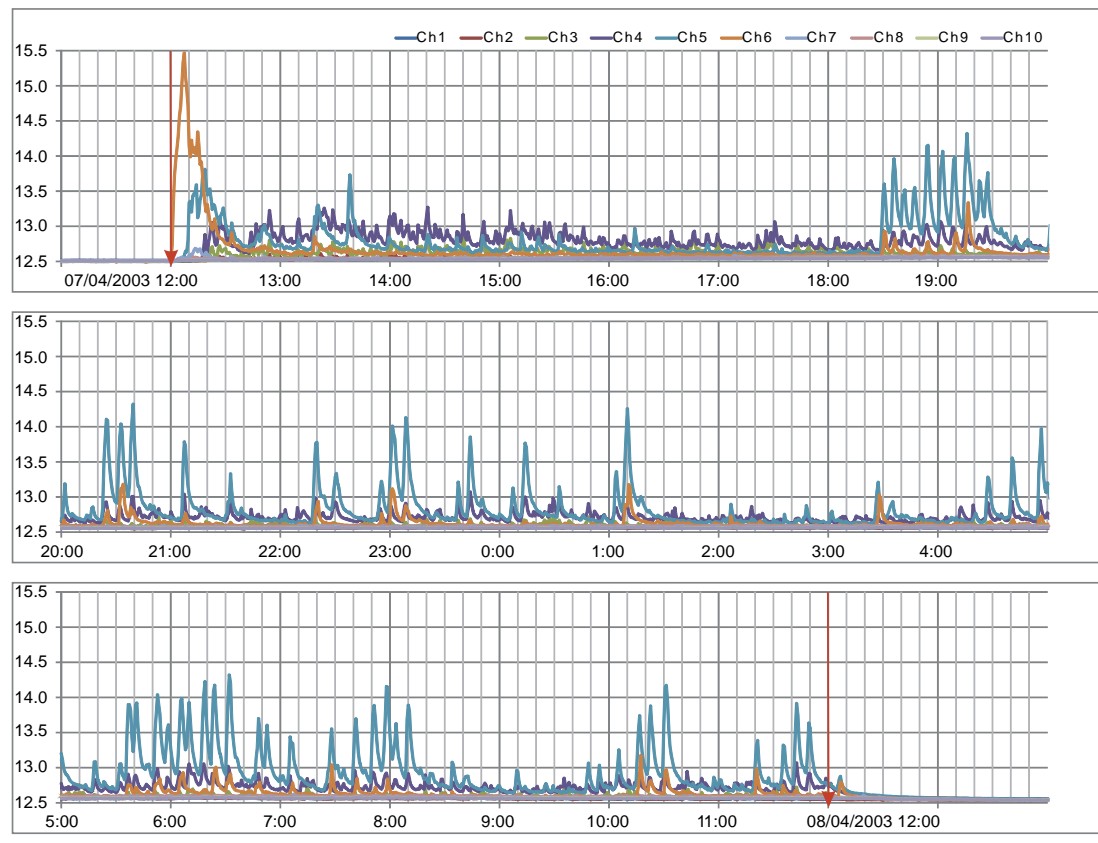

10  **Figure 3: The temperatures in C° at 1-meter height 1 hour before, during, and 2 hours after the A5a experiment.**





**Figure 4: The temperatures 3 hours before, during, and 3 hours after the A5a heating experiment in space-time domain: temperature color-coded in C°; on the vertical axis are shown the location of the sensors on the horizontal 2-m bar; temporal tick marks on horizontal axis correspond to 3600 s = 1 hour.**





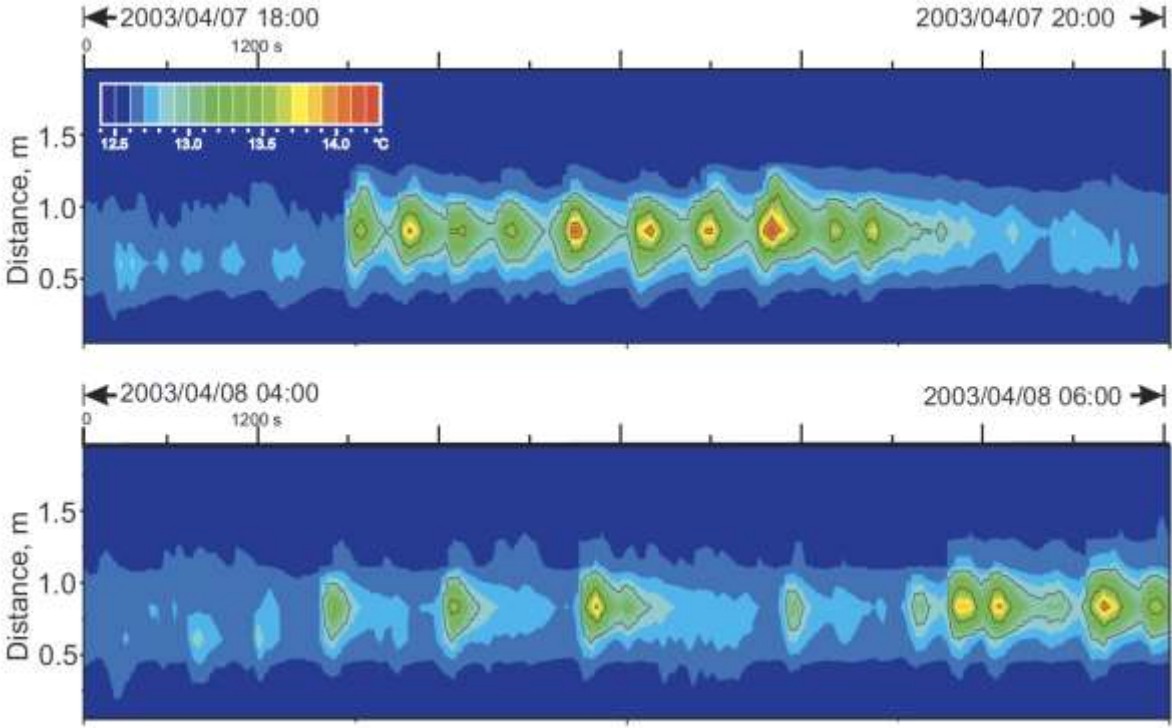

**Figure 5: The temperatures in the space-time domain during two hours of the A5a heating experiment when a long train (upper plate) and a likely "doubling of period" (first half of the lower plate) of strong heat pulses are observed. Times of the beginning and ending are indicated for each plate.**



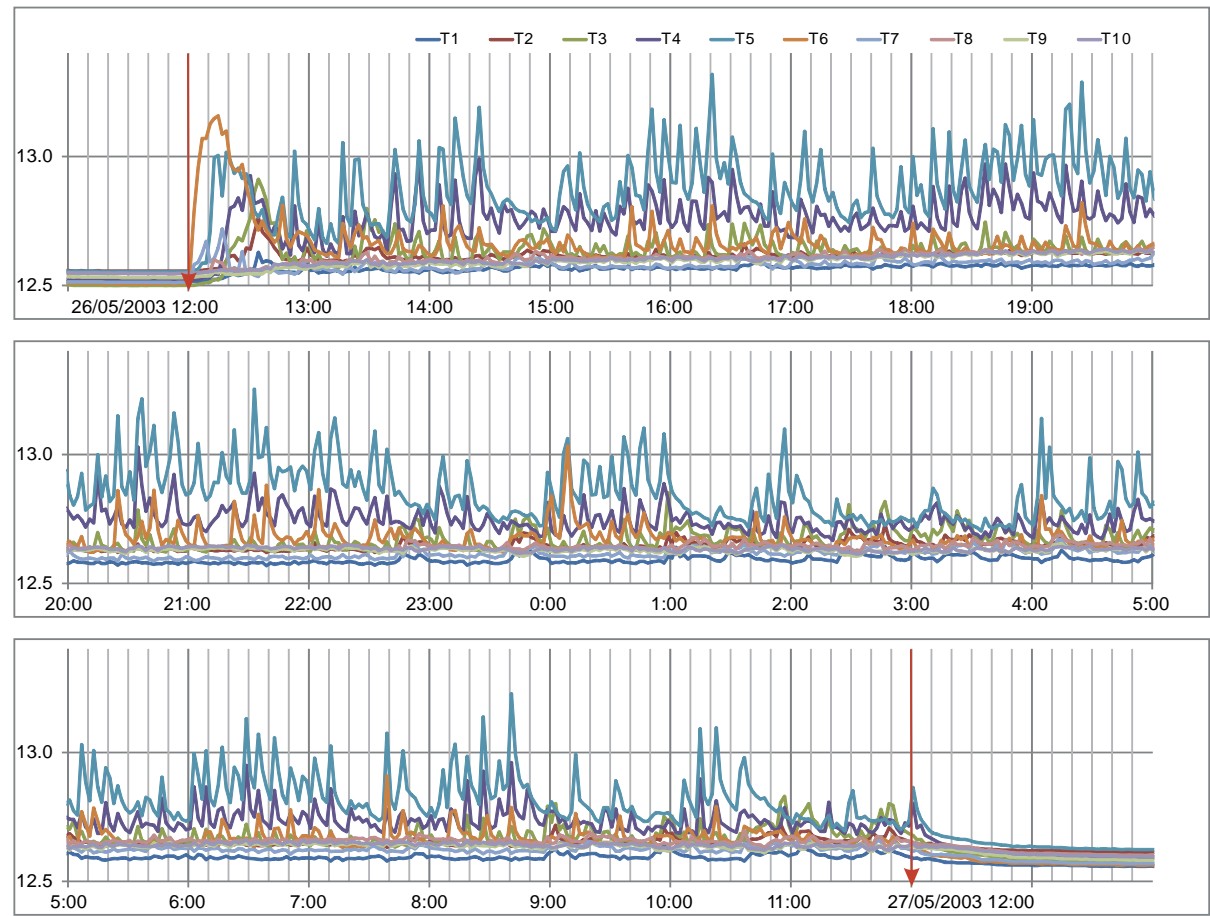

**Figure 6: The temperatures in C° at 1.5-meter height 1 hour before, during, and 2 hours after the A7a experiment.**

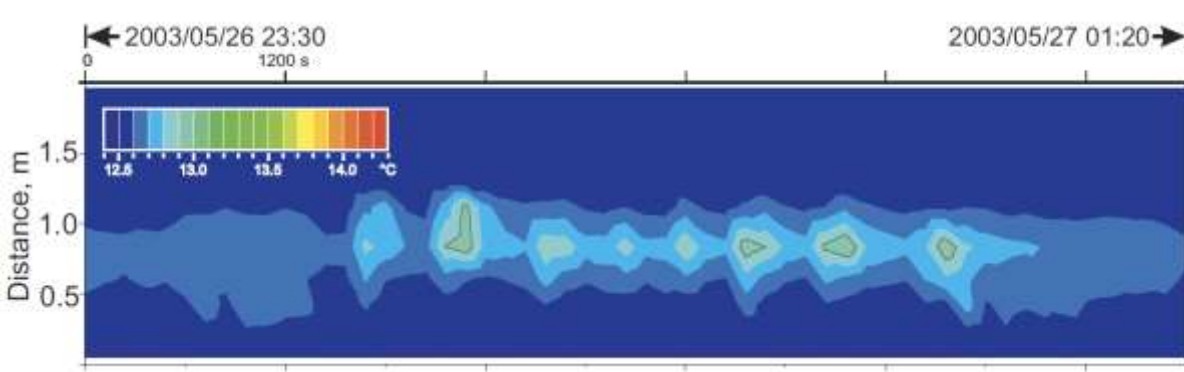

**Figure 7: The temperatures in space-time domain during 110 minutes of the A7a heating experiment where a long "train" of strong heat pulses is observed. The color-coding is the same as in Figs. 4 and 5.**







**Figure 8: The temperatures in C° at 2-meter height 1hour before, during, and 3 hours after the A6a experiment.**





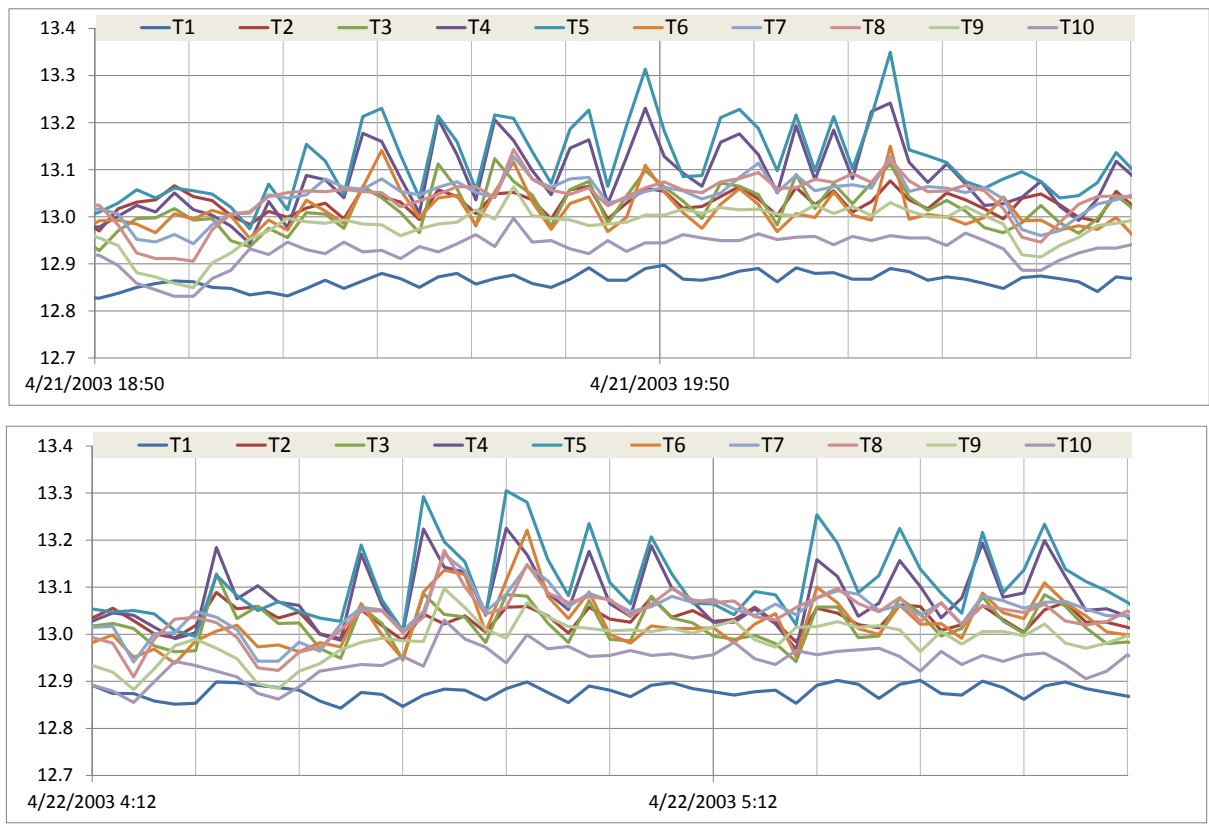

**Figure 9: The temperatures during the A6a experiment heating in the periods 2003/04/21 18:54:52-2003/04/21 (upper plate) and 2003/04/22 04:12:52-2003/04/22 05:52:52 (lower plate): Temperature is given in C°.**

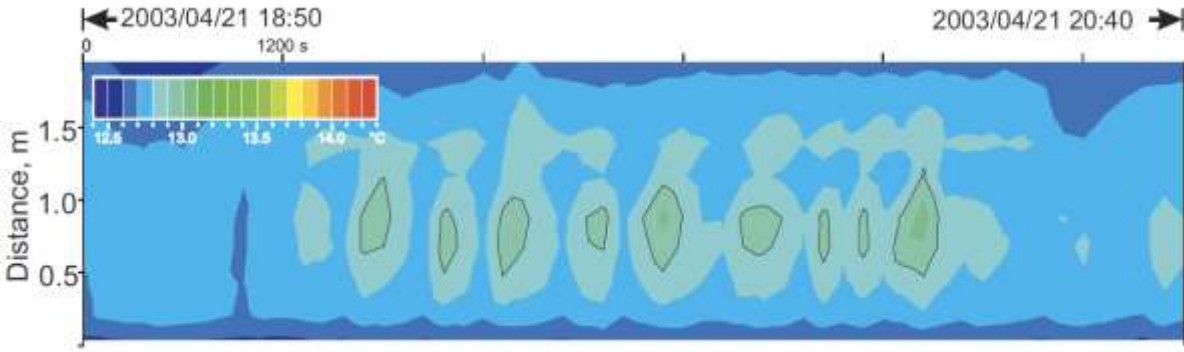

10    **Figure 10: Temperatures in space-time domain during 110 minutes of the A6a heating experiment where a long "train" of strong heat pulses is observed. The colour coding is the same as in Figs. 4 and 5. Note the presence of concomitant pulses aside of the main series.**





**Figure 11: Temperatures before, during, and after the A3 heating experiment in space-time domain: Temperature color-coded in °C; on the vertical axis is represented the position of the sensors on the vertical bar (from 10 at the bottom to 1 on the top).**





a

b

**Figure 12: Temperatures in space-time domain during the three 3-hour intervals (a) and 30 minutes (b) of the A3 heating experiment.**





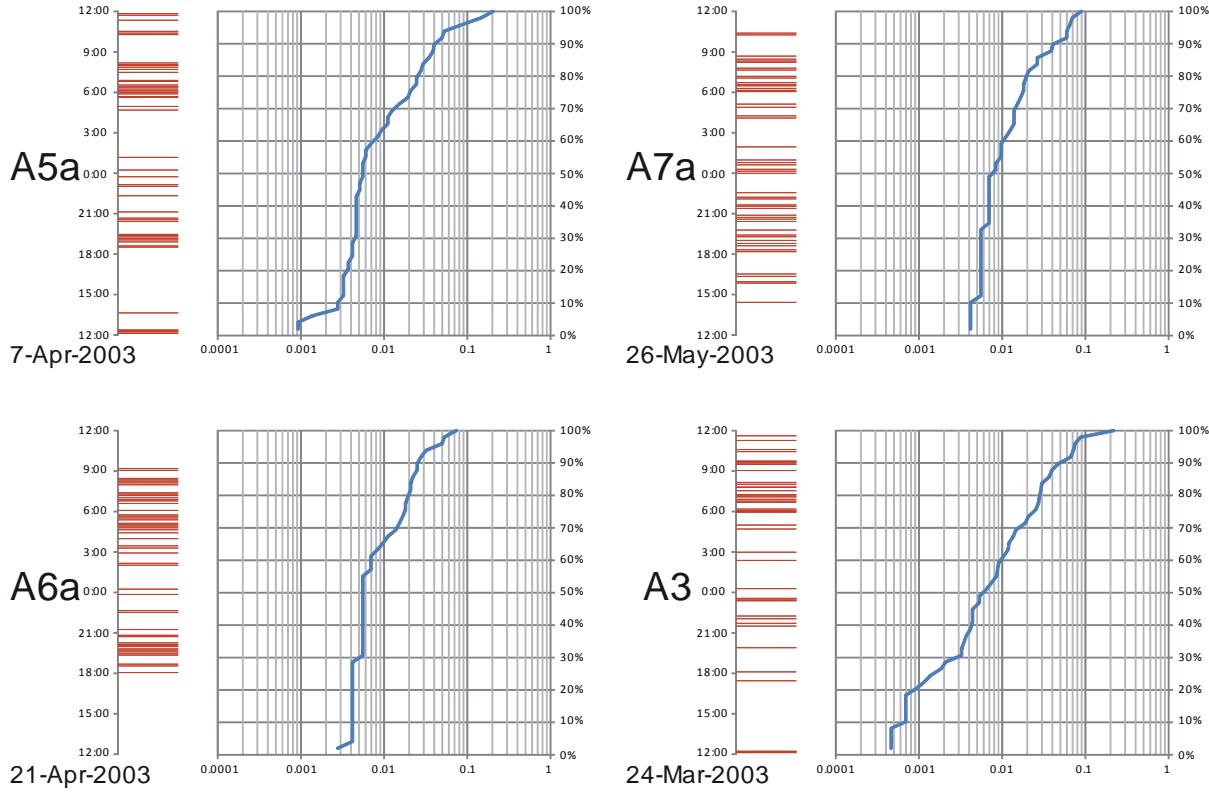

**Figure 13: Times of the 50 largest local maxima of the average temperature of all the sensors on AIII bar (left), and the empirical distribution function of their inter-event times, in units of fractions of a day (right) observed in the A5a, A7a, A6a, and A3 experiments. Time increases when going up.**





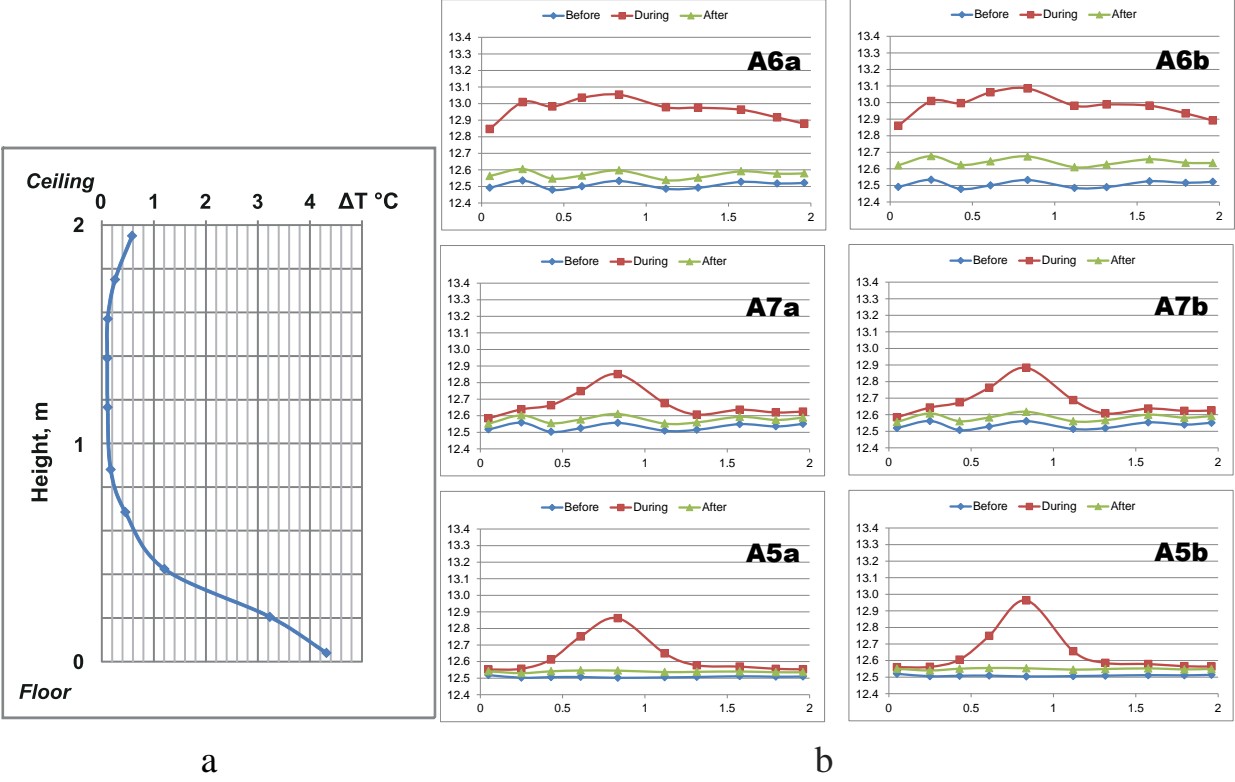

a                b

**Figure 14: (a) The difference $\Delta T = T_h - T_0$ of the average temperatures in 24 hours during ($T_h$) and 12 hours before ($T_0$) a heating experiment with a vertical bar AIII (experiment, A3). (b) The average temperatures on a horizontal bar AIII in 12 hours before (blue), 24 hours during (red), and 12 hours after a heating experiment at 2-m (top), 1.5-m (middle), and 1-m (bottom) heights (experiments A6a, A6b, A7a, A7b, A5a, and A5b, correspondingly).**



**Figure 15: The power spectra, as a function of frequency in Hz, of the first derivative of the average temperature during A6a, A6c (both with the bar at 2 meters, top), A7a, A7b (both with the bar at 1.5 meters, middle), A5a, and A5b (both with the bar at 1 meters, bottom). Each maximum is supplied with the frequency and the power.**





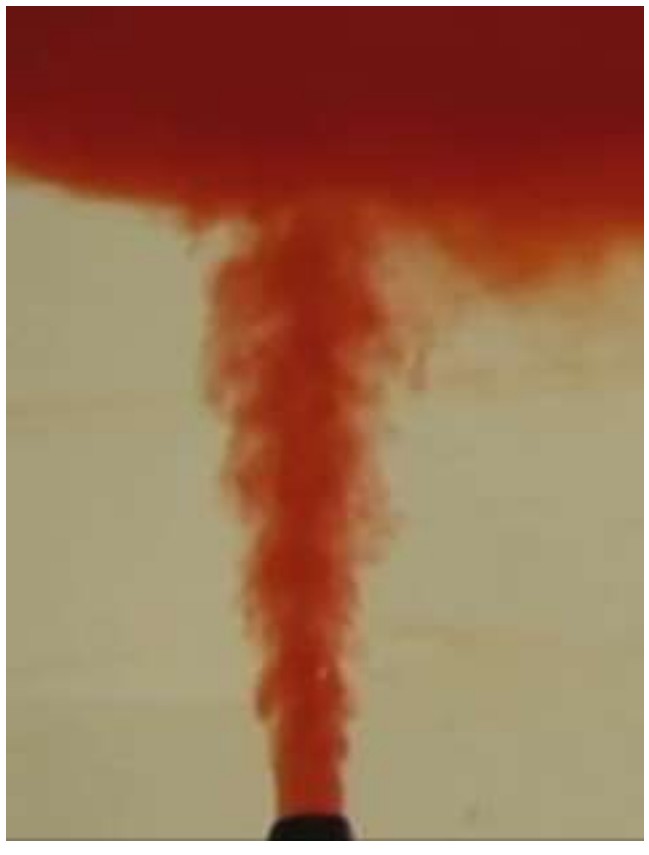

**Figure 16: A snapshot from a movie (Carrozo et al., 2014, Supplementary Material S3), recording an experiment simulating a volcanic vent in a stratified water tank.**

