# Peer review of "Intermittent heat instabilities in an air plume"

_Nonlinear Processes in Geophysics, 2016_

## Referee Comment (RC1) · Anonymous Referee #1 · 22 Apr 2016

**Review of manuscript:**

**Intermittent heat instabilities in an air plume**

Submitted to Non Linear Processes in Geophysics.

Author(s): Jean-Louis Le Mouël[1], Vladimir G. Kossobokov[1,2], Frederic Perrier[1], and Pierre Morat.

MS No.: NPG-2016-23
doi:10.5194/npg-2016-23

**General comments**

This paper presents a novel plume experiment which dimensions are not the typical ones. The presented experiment is very interesting and original because it is between the laboratory and the real geophysical situations. It is clearly stated by the authors that they experimental results agree with the classical model of Morton et al. (1956). The paper studies the heat instabilities of the plume dynamics which behaviour is not well understood and it is an important topic for industrial and geophysical situations. The authors of this work propose to examine the dynamical behaviour of these instabilities and they clearly show that the instabilities have an intermittent character which is presented by means of different kind of figures. The researchers stablish that this dynamics is a universal feature of turbulent plumes in geophysical environments and it could be represented by a smooth spatial function with a global temporal intermittent function. Related to this question, the authors should explain better what is their objective at introduction, at sections 5 and 6 (conclusions). It should be better to introduce a basic description of heat instabilities and intermittency (at introduction). The authors should explain better how their results are important implications for geophysical or industrial situations.

This paper should undergo a minor revision before being considered for publication.

**Specific comments**

**Sub-Section 2.1. The Vincennes quarry**

- **Page 2, Lines 23-24.** What is the reason for the temperature increase of about 0.1ºC per year? Why it is different from the annual variation (0.8 ºC)?
- **Page 2, Lines 29.** You speak about the exceptional conditions of stability. How do you characterize and measure this stability? At this point, I understand that is stable stratified but do you mean that the air of the quarry has a stable stratification or a neutral one?
- **Page 3, Lines 2-5.** You speak about the air exchanges. What is the effect of natural ventilation on stratification of the quarry air?
- Could you also explain in more detail the effect of the quarry walls and ceiling on your experiment and your results?

**Sub-Section 2.3. Configuration of the measurement device of the experiment**

- **Page 4, Lines 2.** This is the first time you mention the screen ("*and that of the screen*"). Therefore, you must explain here what the screen is. This explanation is written in lines 14-15: "*To study the possible contribution of sideway radiations, a screen was placed around the source in 15 experiments A3, A5-A7 (Table 1)*". Eliminate these lines and write this explanation in line 2.
- **Page 4, Line 9.** You write that "*Most of the heating experiments (............) lasted for 24 hours*". Explain how such 24 hours experiments affect on the stratification of the room. Has the heating source of power 100 W been switched on during the 24 hours experiments or only some hours, minutes? The time that the heating source is on, could it have some effect on the plume behaviour?

**Sub-Section 3.1. The horizontal recordings at 1.00 m**

- Sensor 7 is symmetrically placed with respect to the sensor 5. Why their measures are different (figure 3, for example).
- **Page 5, Line 9.** You write "*These heat pulses of instability should not be taken as physical drops of air observed at a given time*". Do you think there is a relation between these pulses and the continuous or intermittent hot air supply (it depends on whether the heat source is turned on all the time).

**Sub-Section 3.5. Joint study of the various time series**

- **Page 6, Line 21.** Why the threshold is 12 minutes?

**Section 4. Averaged temperature profiles**

- **Page 7, Line 1.** Change this sentence "*temperature radial profile*" by the following "*radial temperature profile*".
- **Page 7, Line 13.** Change this sentence.
- **Page 7, Line 17.** How do you calculate the mean of $\Theta$ in the section of the plume at the 1-meter altitude because figure 3b does not exist.
- **Page 7, Line 25.** Eliminate the word "*Remark:*"
- **Page 7, Lines 25 to the end of the section 4.** I think that this paragraph could be placed at the beginning of the section as an objective and to clarify the development of the section 4.

**Section 5. Some global characteristics of the plume dynamics**

- **Page 8, Lines 13-18.** You do not write the values of the sub-index $j$. For me, it is not clear the meaning of the subscript: does $i$ represent the different sensors and $j$ the different heights? If it is so, I do not understand equation (8). Therefore, does $i$ and $j$ represent the different sensors?
- **Page 8, Lines 13-18.** If equation (9) is an extrapolation of equation (8), you say that at every height $z$ it verifies equation (8) and subscript $i$ is not $z$. Therefore, the constant parameters $a_{ij}$ are transformed into functions that depend on the radial coordinate. Could you explain and justify better this extrapolation?

**Section 6. Discussion and conclusions**

- **Page 9, Line 8.** The title of your work is related to instabilities but it is necessary to speak more about it. Could you explain better what kind of

instabilities (already known or new ones) are you studying? What aspects of these instabilities are described and clarified by your experiments?

- Do you think there is some relationship between the instabilities you see/measure and the meandering phenomenon, which appears in geophysical situations under stable conditions?
- **Page 9, Line 30-32.** One of their major contributions is to propose a first order factorization of the spatial and temporal variations. Highlight this aspect and develop it.

**Technical comments**

- Use units in seconds, not minutes.

**Section 1. Introduction.**

- **Page 2, Lines 4 to 7.** The text is repeated, it is just the same that at the beginning of the Introduction. You have to eliminate lines 4 to 7.

**Section 2.1. Vincennes quarry.**
- It woulb be interesting to add a photograph of the quarry (in Figure 1).

**Section 2.2. Temperature measurements. Thermistors. Calibration.**
- Change the name of the section (use a phrase, for example "Temperature measurements by thermistors").
- **Page 3, Line 18.** Change "section II.3" to "section 2.3" because your notation is "2.3 Configuration of the measurement…..".
- **Page 3, Line 16**. Rewrite the expressión

$$T^1 = A + B\ lnR + C\ (lnR)^2,$$

because the superindex of $T$ is not clear.

**Section 3. Results: Temperature fluctuations in the plume.**
- **Page 4, Line 18.** Eliminate the notation "ch1, ch2, … ,ch10" because you do not use it again. Write "by 10 sensors" and add a reference to a figure.
- **Page 4, Line 18.** Why do not you order the names of the experiments: A5a, A5b, A6a, A6b, A7a and A7b? Why do not you mention experiment A4?

**Section 3.1. The horizontal recordings at 1.00 m.**
- **Page 4, Line 27.** You mention that "Figure 3 shows the …temperature …..from 12:00…to 16:00.." but, really, is 14:00.

**Section 3.2. The horizontal recordings at 1.50 m.**
- Change "section III.1" to "section 3.1" because your notation is "3.1 The horizontal recordings at 1.00 m".

**Section 3.4. AIII device maintained vertically.**
- **Page 6, Line 10-11.** Eliminate the definition of $t_k$ ($t_k=t_0+20k$) in line 11 and write it in line 10.

**Section 3.5. Joint study of the various time series.**
- **Page 6, Line 18.** Order the names of the experiments as (A5a, A6a, A7a)

**Section 4. Averaged temperature profiles.**
- **Page 7, Lines 5-6.** Are necessary the quotation marks at the following phrase: *"No velocity scale is provided by the specification of a free convection situation"*.
- **Page 7, Line 9.** Change the notation of the upward vertical unit vector to **k.**
- **Page 7, Lines 17.** Change "m$^2$; from" to "m$^2$. From".
- **Page 7, Lines 17.** Figure 3b does not exist.

**Section 5. Some global characteristics of the plume dynamics**
- **Page 8, Line 4.** Change "section IV" to "section 4" because your notation is "4. Averaged temperature profiles".
- **Page 8, Line 18.** Rewrite the sentence. Write "*$a_{ij}$ being (constants) parameters*" at the beginning of the phrase. Change the sentence "*taking j = 5 of the sensor seeing the large variations*".
- **Page 8, Line 19.** Why do you change the coefficient $a_{i5}$ to *b*?
- **Page 8, Line 22.** Eliminate the phrase *"en bloc"* and rewrite the phrase.

**Section 6 Discussion and conclusions**
- **Page 10, Line 4.** Change the reference *(Hernandez et al., 2015)* to *(Hernandez, 2015)*.

**References**

Revise all the bibliographic refences because the following references are not cited in the text:

- *Carazzo, G., Girault, F., Aubry, T., Bouquerel, H., and Kaminski, E.: Laboratory experiments of forced plumes in a density stratified crossflow and implications for volcanic plumes, Geoph. Res. Lett. **41**: 8759–8766, 2014.*
- *Carazzo, G., Kaminski, E., and Tait S.: The route to self-similarity in turbulent jets and plumes. J. Fluid Mech. **547**: 137–148, 2006.*
- *Fischer, H. B., Imberger, J., List, E. J., Koh, R. C. Y., and Brooks N. H.: Mixing in inland and coastal waters. Academia Press, New York, United States. 483 p., 1979.*
- *George, W. K. Jr., Alpert, R. L., and Tamanini, F.: Turbulence measurements in an axisymmetric buoyant plume. Int. J. Heat Mass Transfer **20**(11): 1145–1154, 1977.*
- *Guyon, E., Hulin, J.-P., Petit, L., and Mitescu, C.D.: Physical Hydrodymanics (2nd ed.). Oxford University Press, Oxford, 536 p., 2015.*
- Kaminski, E., Tait, S., and Carazzo, G.: Turbulent entrainment in jets with arbitrary buoyancy. *J. Fluid Mech.* **526**: 361–376, 2005.
- Morat, P., Le Mouël, J.-L., Poirier, J.-P., and Kossobokov, V.: Heat and water transport by oscillatory convection in an underground cavity. *C. R. Acad. Sci. Paris* **328** (1): 1–8, 1999.
- Tritton, D. J.: *Physical Fluid Dynamics*. Clarendon Press, 544 p, 1988.

You mention the following reference (Crouzeix et al., 2006) in the text but you write three different papers which have the same reference:

- *Crouzeix, C., Le Mouël, J.-L., Perrier, F., Shnirman, M. G., and Blanter E.: Long-term persistence of the spatial organization of temperature fluctuation lifetime in turbulent air avalanches. Phys. Rev. E **74**, 036308, 2006. Are these the right page?*

- *Crouzeix, C., Le Mouël, J.-L., Perrier, F., and Richon, P.: Non-adiabatic boundaries and thermal stratification in a confined volume. Int. J. Heat Mass Transfer **49**: 1974-1980, 2006.*
- *Crouzeix, C., Le Mouël, J.-L., Perrier, F., and Richon, P.: Thermal stratification induced by heating in a non-adiabatic context. Building and Environment **41** (7): 926–939, 2006.*

If you mention the three papers in the text, distinguish them with different references, something like (Carazzo et al., 2006(a), (b), (c)).

**Tables**
- **Table 1.** Explain the meaning of the signs - and + related to source raised and screen in the table caption.
- **Table 2.** Express all in seconds, not in minutes, as in Table 1.
- **Table 2.** The meaning of the column "*max(Δt), s*" is not explained in the table caption.

**Figures**
- **Figure 2.** Explain the meaning of the numbers "1, 2, 3, …,10" in the table caption. Also explain what is the screen and the meaning of AI, AII, AIII and S. The reason is that when figure 2 is mentioned, these concepts have not yet been explained.
- **Figure 3.** Explain the meaning of "Ch1, Ch2,…Ch10" in the table caption.
- **Figure 6.** Explain the meaning of "T1, T2,…T10" in the table caption. Is T1 the same instrument that Ch1, and so on? If they are, use the same notation in all figures: Ch1 or T1.
- **Figure 8 and 9.** As in Figure 6.
- **Figure 9.** Rewrite the information related to dates and hours (*the periods 2003/04/21 18:54:52-2003/04/21 (upper plate) and 2003/04/22 04:12:52-2003/04/22 05:52:52*) because it is a bit confusing.
- **Figure 13.** Change "*A5a, A7a, A6a and A3 experiments*" to "*A3, A5a, A6a and A7a experiments*".

---

## Referee Comment (RC2) · Anonymous Referee #1 · 1 Jun 2016

Good morning.

I have revised the reply of the authors of the paper entitled "Intermittent heat instabilities in an air plume" and they have done all the corrections and I agree with their explanations and suggestions. Therefore, the article is prepared to proceed with editorial process.

---

## Referee Comment (RC3) · Anonymous Referee #2 · 14 Jun 2016

The manuscript present an interesting experimental investigation of thermal plumes in a controlled environment. The most peculiar novelty of this work is the location of the experiment, which is in a naturally thermally stable room of an abandoned underground quarry. The authors performed a quite exhaustive series of measurements to characterize the properties of the thermal plumes generated by one source. The description of the experiment is quite clear and concise, and the results drawn by the authors are described in terms of temporal intermittency and its relationship with the instabilities arising in their set-up. The article is interesting and deserve publication, although a more thoughtful interpretation of the results in terms of physics of the system could be beneficial.

I have a few minor comments, which will be listed below.

* lines 26-29 and 41-45: this sentence is repeated, please remove the second one.

[Figure]

* line 89: there is an extra space in the title II.2: "C alibration"

* page 5: the authors should probably describe the stability of the source in terms of temperature variation, and presence of possible periodicity, which could affect their results. This is perhaps a major point that should be described with care in the text, and possibly tested in the lab.

* Table 1, the entry in the lower-right case is missing the unit of measure.

* lines 122-123: this description perhaps belongs to previous section?

* line 130: what is the set-up S? It only appears here and should be commented.

* line 134: the experiments last for 24 hours. Did the authors test at least in one case what happens for longer measurements?

* lines 145-147: the lists of experiments should be sorted in alpha-numerical order (in the present version, A7 comes before A6), unless there is a reason for this order, and in this case it should be clearly stated.

* line 159: Do the authors have any explaination for the difference between probes 4, 5 and 6? Is it depending on the position? Why?

* line 204: there are two periods at the end of the sentence: "..".

* page 15, section III.4: could the authors comment about their findings in this configuration?

* line 259: the authors show the cumulative distribution.

* line 304: the parameter UL/nu is the Reynolds number. The auhtors should acknowledge that and perhaps comment in terms of laminarity/turbulence of the flow, as resulting from their estimate $10^4$?

* line 317: What is the consequence of the satisfactory agreement between F and F_0? Could the authors comment on that more explicitly in the text?

* lines 332-336: did the authors look at the log-log plot of the spectra' does it present a power-law? This could be an interesting information to add to the article, even to describe the turbulence properties of the system (and given the high Reynolds number, I suspect it should be a power-law)...

* page 7: the authors speak about instabilities, but they do not really refer or describe what particular type of instability is relevant here. Could they please spend some words on this?

---

## Author Response (AR1)

**Answers to Reviewer #1**

We thank the reviewer for in detail study of our manuscript and providing comments and valuable remarks that help improving presentation of the results of the heat experiments in an isolated room of the Vincent quarry. Our responses to each of the reviewer's comments are given in red through the text of his or her **Review**.

**Review of manuscript:**
**Intermittent heat instabilities in an air plume**

Submitted to Non Linear Processes in Geophysics.
Author(s): Jean-Louis Le Mouël[1], Vladimir G. Kossobokov[1,2], Frederic Perrier[1], and Pierre Morat.
MS No.: NPG-2016-23
doi:10.5194/npg-2016-23

**General comments**

This paper presents a novel plume experiment which dimensions are not the typical ones. The presented experiment is very interesting and original because it is between the laboratory and the real geophysical situations. It is clearly stated by the authors that they experimental results agree with the classical model of Morton et al. (1956). The paper studies the heat instabilities of the plume dynamics which behaviour is not well understood and it is an important topic for industrial and geophysical situations. The authors of this work propose to examine the dynamical behaviour of these instabilities and they clearly show that the instabilities have an intermittent character which is presented by means of different kind of figures. The researchers stablish that this dynamics is a universal feature of turbulent plumes in geophysical environments and it could be represented by a smooth spatial function with a global temporal intermittent function. Related to this question, the authors should explain better what is their objective at introduction, at sections 5 and 6 (conclusions). It should be better to introduce a basic description of heat instabilities and intermittency (at introduction). The authors should explain better how their results are important implications for geophysical or industrial situations.
This paper should undergo a minor revision before being considered for publication.

We added a paragraph at the end of Introduction, which was originally excluded for sake of shortening the size of the manuscript.

**Specific comments**

**Sub-Section 2.1. The Vincennes quarry**
☐**Page 2, Lines 23-24.** What is the reason for the temperature increase of about 0.1℃ per year? Why it is different from the annual variation (0.8 ℃)?

The paragraph refers to the estimations obtained in the earlier studies that are well cited. The annual average temperature in the quarry increased from 12°C in 2003 to 12.2°C in 2005 which change is well in range of the local change of the annual average temperature in Paris (13.2°C in 2003, 13.0°C in 2004, and 12.9°C in 2005), while the annual variance of the temperature remained about the same (i.e., 0.8°C). To avoid any misinterpretation we have changed the text to the following –

> Temperature had been measured in the quarry since 2001, giving in 2003-2005 an annual mean temperature in range from 12°C to 12.2°C with a seasonal variation of the order of 0.8°C related to natural ventilation through the access pit (Perrier et al., 2004; Perrier and Richon, 2010).

☐**Page 2, Lines 29.** You speak about the exceptional conditions of stability. How do you characterize and measure this stability? At this point, I understand that is stable stratified but do you mean that the air of the quarry has a stable stratification or a neutral one?

The preceding paragraph demonstrates quantitatively how stable are temperatures and humidity in the abandoned quarry as a whole. The conditions in the room S15 which is far away of the entrance to the quarry are even more stable as can be judged from the temperature measurements at the times of no visits and of those not affected by heating experiments (i.e. from Wednesday to Sunday) and estimated ventilation rate ("of the order of $2 \times 10^{-6}$ s$^{-1}$ ").

☐**Page 3, Lines 2-5.** You speak about the air exchanges. What is the effect of natural ventilation on stratification of the quarry air?

☐ Could you also explain in more detail the effect of the quarry walls and ceiling on your experiment and your results?

We presume that the air exchanges in room S15 with such a low ventilation rate are negligible in relation to the natural stratification of the air in its volume of about 400 m$^3$. Having in mind the location and dimensions of resulting plume in our heating experiments we may also consider negligible the effect of the distant walls, while the ceiling is acting as the limiting boundary in the system. Naturally, the ceiling, walls, and the floor act as cooling elements of the system where hot air rising from the heat source is spreading through the volume of the room.- This is added at the end of section 2.1.

**Sub-Section 2.3. Configuration of the measurement device of the experiment**
☐**Page 4, Lines 2.** This is the first time you mention the screen ("*and that of the screen*"). Therefore, you must explain here what the screen is. This explanation is written in lines 14-15: "*To study the possible contribution of sideway radiations, a screen was placed around the source in 15 experiments A3, A5-A7 (Table 1)*". Eliminate these lines and write this explanation in line 2.

Done.

☐**Page 4, Line 9.** You write that "*Most of the heating experiments (………..) lasted for 24 hours*". Explain how such 24 hours experiments affect on the stratification of the room. Has the heating source of power 100 W been switched on during the 24 hours experiments or only some hours, minutes? The time that the heating source is on, could it have some effect on the plume behaviour?

We added "when the heat source was switched on" and "when the heat source was switched off" for a clarification.

**Sub-Section 3.1. The horizontal recordings at 1.00 m**

☐ Sensor 7 is symmetrically placed with respect to the sensor 5. Why their measures are different (figure 3, for example).

We are sorry, but it is hard observing any symmetry in between sensors 5 and 7. Yes, you are right, at 1 m height sensor 6 was the first affected with the rising hot air, but in about 6 hours the axis of the plume had developed next to sensor 5 as can be judged from the A5 heating experiments at 1 m height (Fig. 3 shows A5a, and A5b looks very similar). One should try finding some symmetry between sensors 4 and 6, 3 and 7, etc., as clearly confirmed by space-time diagrams in Figs. 4 and 5.

☐**Page 5, Line 9.** You write "*These heat pulses of instability should not be taken as physical drops of air observed at a given time*". Do you think there is a relation between these pulses and the continuous or intermittent hot air supply (it depends on whether the heat source is turned on all the time).

Yes, the heat source was turned on during all the 24 hours in each experiment, and we believe that an intermittent pattern of heat pulses and their absence is due to a non-linear effect involving stream of hot air in a stratified environment.

**Sub-Section 3.5. Joint study of the various time series**

☐**Page 6, Line 21.** Why the threshold is 12 minutes?

Yes, 12 minutes = 720 s is an arbitrary choice of the threshold that determines a classification into short and long intervals between heat pulses. Some motivation could be related to the selected sampling interval of 120 s = 2 minutes in the experiments A6 and A7 or to doubling of the typical decay time of one pulse ~360 s, as well as to the change in the slope of the empirical distribution functions given in Fig. 13.

**Section 4. Averaged temperature profiles**

☐**Page 7, Line 1.** Change this sentence "*temperature radial profile*" by the following "*radial temperature profile*".

Done.

☐**Page 7, Line 13.** Change this sentence.

Done.

☐**Page 7, Line 17.** How do you calculate the mean of $\Theta$ in the section of the plume at the 1-meter altitude because figure 3b does not exist.

Thanks for pointing to the wrong reference, which should be Fig. 14b. Corrected.

☐**Page 7, Line 25.** Eliminate the word "*Remark:*"

Done.

☐**Page 7, Lines 25 to the end of the section 4.** I think that this paragraph could be placed at the beginning of the section as an objective and to clarify the development of the section 4.

Thanks. Following the reviewer's suggestion the paragraph is moved to the beginning of Section 4.

**Section 5. Some global characteristics of the plume dynamics**
☐**Page 8, Lines 13-18.** You do not write the values of the sub-index $j$. For me, it is not clear the meaning of the subscript: does $i$ represent the different sensors and $j$ the different heights? If it is so, I do not understand equation (8). Therefore, does $i$ and $j$ represent the different sensors?

Yes, it does; $i$ and $j$ do represent different sensors. Thanks for pointing to the typos. Following the reviewer's comment we rephrased and corrected the paragraph.

☐**Page 8, Lines 13-18.** If equation (9) is an extrapolation of equation (8), you say that at every height $z$ it verifies equation (8) and subscript $i$ is not $z$. Therefore, the constant parameters $a_{ij}$ are transformed into functions that depend on the radial coordinate. Could you explain and justify better this extrapolation?

Yes, $i$ is not $z$ in (9); naturally, discreet $i$ at the height $z$ is transformed into continuous $r$.

**Section 6. Discussion and conclusions**
☐**Page 9, Line 8.** The title of your work is related to instabilities but it is necessary to speak more about it. Could you explain better what kind of instabilities (already known or new ones) are you studying? What aspects of these instabilities are described and clarified by your experiments?
☐ Do you think there is some relationship between the instabilities you see/measure and the meandering phenomenon, which appears in geophysical situations under stable conditions?

Following the reviewer's suggestion we have clarified what kind of instability appears in the heating experiments in the preceding paragraph. We believe that the meandering phenomenon can be ruled out due to rather fast speed of the air flow one can observe in Fig. 12 representing the dynamical propagation of the heat pulses in vertical direction.

☐**Page 9, Line 30-32.** One of their major contributions is to propose a first order factorization of the spatial and temporal variations. Highlight this aspect and develop it.

Following the reviewer's suggestion, the assumption of a first order factorization in the models is highlighted.

**Technical comments**

☐ Use units in seconds, not minutes.
 Following the reviewer's comment we supplied (where appropriate) the time measurements in minutes with the equivalent values in seconds.

**Section 1. Introduction.**
☐**Page 2, Lines 4 to 7.** The text is repeated, it is just the same that at the beginning of the Introduction. You have to eliminate lines 4 to 7.
Done.

**Section 2.1. Vincennes quarry.**
☐ It woulb be interesting to add a photograph of the quarry (in Figure 1).
We have photos of the room S15 with the heating experiment set up (e.g. the one below) and can provide it, if the editor finds it appropriate and necessary.

[Figure]

**Section 2.2. Temperature measurements. Thermistors. Calibration.**
☐ Change the name of the section (use a phrase, for example "Temperature measurements by thermistors").
Done.

☐**Page 3, Line 18.** Change "section II.3" to "section 2.3" because your notation is "2.3 Configuration of the measurement…..".
 Done.

☐**Page 3, Line 16**. Rewrite the expressión because the superindex of $T$ is not clear.
Done.

**Section 3. Results: Temperature fluctuations in the plume.**
☐**Page 4, Line 18.** Eliminate the notation "ch1, ch2, … ,ch10" because you do not use it again. Write "by 10 sensors" and add a reference to a figure.

Done.

☐**Page 4, Line 18.** Why do not you order the names of the experiments: A5a, A5b, A6a, A6b, A7a and A7b? Why do not you mention experiment A4?

Unlike the experiments A5, A6, and A7, the heat source was not surrounded by the screen.

**Section 3.1. The horizontal recordings at 1.00 m.**
☐**Page 4, Line 27.** You mention that "Figure 3 shows the …temperature …..from 12:00…to 16:00.." but, really, is 14:00.

Thanks. Changed to the correct interval "…from 11:00 … to 14:00…".

**Section 3.2. The horizontal recordings at 1.50 m.**
☐ Change "section III.1" to "section 3.1" because your notation is "3.1 The horizontal recordings at 1.00 m".

Done.

**Section 3.4. AIII device maintained vertically.**
☐**Page 6, Line 10-11.** Eliminate the definition of $t_k$ ($t_k=t_0+20k$) in line 11 and write it in line 10.

Done.

**Section 3.5. Joint study of the various time series.**
☐**Page 6, Line 18.** Order the names of the experiments as (A5a, A6a, A7a)

The order in the text corresponds to the order by rising heights in the same sentence.

**Section 4. Averaged temperature profiles.**
☐**Page 7, Lines 5-6.** Are necessary the quotation marks at the following phrase: *"No velocity scale is provided by the specification of a free convection situation"*.

We believe it is necessary as being cited from (Morton et al., 1956).

☐**Page 7, Line 9.** Change the notation of the upward vertical unit vector to **k**.

Sorry, we wish preserving the notation.

☐**Page 7, Lines 17.** Change "$m_2$; from" to "$m_2$. From".

Done.

☐**Page 7, Lines 17.** Figure 3b does not exist.

Changed for 14b.

**Section 5. Some global characteristics of the plume dynamics**
☐**Page 8, Line 4.** Change "section IV" to "section 4" because your notation is "4. Averaged temperature profiles".

Done.

☐**Page 8, Line 18.** Rewrite the sentence. Write "*$a_{ij}$ being (constants) parameters*" at the beginning of the phrase. Change the sentence "*taking j = 5 of the sensor seeing the large variations*".

Done.

☐**Page 8, Line 19.** Why do you change the coefficient $a_{i5}$ to $b$?

Done to get rid of the index 5, so that $b_i$ represents the value of $f_z(r)$ at the position of sensor $i$.

☐**Page 8, Line 22.** Eliminate the phrase *"en bloc"* and rewrite the phrase.

Done.

**Section 6 Discussion and conclusions**
☐**Page 10, Line 4.** Change the reference *(Hernandez et al., 2015)* to *(Hernandez, 2015)*.

Done.

**Tables**
☐**Table 1.** Explain the meaning of the signs - and + related to source raised and screen in the table caption.
Done.
☐**Table 2.** Express all in seconds, not in minutes, as in Table 1.
Done.
☐**Table 2.** The meaning of the column "*max(☐), s*" is not explained in the table caption.
Done.

**Figures**
☐**Figure 2.** Explain the meaning of the numbers "1, 2, 3, …,10" in the table caption. Also explain what is the screen and the meaning of AI, AII, AIII and S. The reason is that when figure 2 is mentioned, these concepts have not yet been explained.
Done. Figure caption is changed accordingly.

☐**Figure 3.** Explain the meaning of "Ch1, Ch2,…Ch10" in the table caption.
Done. Figure and its caption are changed accordingly, so that T1, T2, … , T10 be the same notation in Figs. 3, 6, 8, and 9:
"T1, T2, … , T10 denote temperature series of the ten sensors on the AIII set-up."

☐**Figure 6.** Explain the meaning of "T1, T2,…T10" in the table caption. Is T1 the same instrument that Ch1, and so on? If they are, use the same notation in all figures: Ch1 or T1.
Done.

☐**Figure 8 and 9.** As in Figure 6.
Done.

☐**Figure 9.** Rewrite the information related to dates and hours (*the periods 2003/04/21 18:54:52-2003/04/21 (upper plate) and 2003/04/22 04:12:52-2003/04/22 05:52:52*) because it is a bit confusing.
Thanks for pointing to another typo. Corrected.

☐**Figure 13.** Change "*A5a, A7a, A6a and A3 experiments*" to "*A3, A5a, A6a and A7a experiments*".
Sorry, we wish preserving the order reflecting the height of the AIII set-up.

**Answers to Reviewer #2**

We thank the reviewer for accurate comments and valuable remarks, as well as for raising interesting points on the underlying physics. Our responses to each of the reviewer's comments are given in red through the text of his or her **Review**.
The manuscript present an interesting experimental investigation of thermal plumes in a controlled environment. The most peculiar novelty of this work is the location of the experiment, which is in a naturally thermally stable room of an abandoned underground quarry. The authors performed a quite exhaustive series of measurements to characterize the properties of the thermal plumes generated by one source. The description of the experiment is quite clear and concise, and the results drawn by the authors are described in terms of temporal intermittency and its relationship with the instabilities arising in their set-up. The article is interesting and deserve publication, although a more thoughtful interpretation of the results in terms of physics of the system could be beneficial. Thanks.

I have a few minor comments, which will be listed below.

* lines 26-29 and 41-45: this sentence is repeated, please remove the second one. Done.

* **line 89: there is an extra space in the title II.2: "C alibration"** Following the R#1 suggestion the title has been changed to "**2.2 Temperature measurements by calibrated thermistors**"

* page 5: the authors should probably describe the stability of the source in terms of temperature variation, and presence of possible periodicity, which could affect their results. This is perhaps a major point that should be described with care in the text, and possibly tested in the lab. Dedicated heating experiments were performed in the last stage of the experiments in 2005 with thermistors attached to the source. Temperature of the source does not show any periodic variations in the frequency range of the instabilities mentioned in this paper. This point is now explicitly mentioned in the text at the end of section 2.3 of the revised version.

* Table 1, the entry in the lower-right case is missing the unit of measure. Done.

* lines 122-123: this description perhaps belongs to previous section? Yes, the description of thermistors in the room at distance from the entrance to the quarry may go to the previous rather general section on calibrated thermistors. However, we prefer placing it into the section describing the specifics of installation of the set-ups in the room of heating experiments.

* line 130: what is the set-up S? It only appears here and should be commented. We have changed this part of description of the set-ups: "…while the sensors of set-up S used for the same purpose were placed at about 1.5 cm inside the rock…"

\* line 134: the experiments last for 24 hours. Did the authors test at least in one case what happens for longer measurements? No. Each of the 24-hour heating experiments demonstrates a persistent behavior of the temperature measurements after just a few hours at the very beginning; therefore we believe that the observed behavior would not change in case of an experiment extended for a few hours more.

\* lines 145-147: the lists of experiments should be sorted in alpha-numerical order (in the present version, A7 comes before A6), unless there is a reason for this order, and in this case it should be clearly stated. Yes, there is a reason (i.e. the order by the height of the AIII set-up in horizontal position) which is clearly stated in the revision.

\* line 159: Do the authors have any explanation for the difference between probes 4, 5 and 6? Is it depending on the position? Why? Yes, an obvious one would be the distance from the sensor to the axis of the plume that originates in a heating experiment. Yes, it must depend on the position of a sensor relative to the plume axis, which may and as evident from Fig. 4 does move in time, perhaps, due to natural variability of the environment.

\* line 204: there are two periods at the end of the sentence: "..". Corrected.

\* page 15, section III.4: could the authors comment about their findings in this configuration? As suggested by the reviewer, we added the following comments to the end of section 3.4:

As is evident from Fig.11, in the first 4 minutes after switching on the heat source at 12:00, the temperature T10, i.e. the nearest to the source, is rising slower, while the temperatures T6-T8 grow faster than at any of the other thermistors. At 12:09, the temperature T9 surpasses T6 and T7-T9 rise to their maximal values about 17-18°C. At this moment of time T10 continues to grow steadily, while temperatures T7-T9 start falling down. At 12:16 T10 rise above all the others and by 12:20 it is 1.4-2.6°C higher than any of T1-T9. It appears that at about 12:30 the formation of a plume proceeds to its final stage lasting for about 30 minutes followed by rather regular dynamics with domination of T10 ~16.5-17°C and T9 ~15.4-16°C. From 13:00 on, and till the end of heating experiment, the average values of T10 and T9 are 16.9°C with σ = 0.2°C and 15.8°C with σ = 0.7°C, respectively. The difference in σ's allows for sporadic rise of T9 above T10, with cases are exemplified in Fig. 12. In particular, one can clearly observe rather quick propagation of the heat pulse through the entire plume from the floor to the ceiling of the room S15 (Fig. 12b).

\* line 259: the authors show the cumulative distribution. Yes, we do. The attribute "cumulative" is added in the revision (although assumed by default and is evident from the figure).

\* line 304: the parameter UL/nu is the Reynolds number. The auhtors should acknowledge that and perhaps comment in terms of laminarity/turbulence of the flow, as resulting from their estimate 10^4? As suggested by the reviewer, one sentence has been added to acknowledge the value of UL/v.

\* line 317: What is the consequence of the satisfactory agreement between F and F_0? Could the authors comment on that more explicitly in the text? As suggested by the reviewer, we have added in the revised version a comment elaborating on the agreement between the two values.

\* lines 332-336: did the authors look at the log-log plot of the spectra' does it present a power-law? This could be an interesting information to add to the article, even to describe the turbulence properties of the system (and given the high

Reynolds number, I suspect it should be a power-law)... Of course, we did have a look at the log-log plot of the spectra, which indeed display a complex set of power-law segments in the low frequency domain. We believe that providing this information would distract the reader from the main aim of over article, i.e. intermittent pattern in dynamics in the frequency domain from 120 to 1000 s. You may guess that we plan returning to it in a different paper.

* page 7: the authors speak about instabilities, but they do not really refer or describe what particular type of instability is relevant here. Could they please spend some words on this? Following the reviewer's suggestion we have clarified what kind of instability appears in the heating experiments in our study.

[revised manuscript text omitted]